# The Role of Tetrahydrocurcumin in Tumor and Neurodegenerative Diseases Through Anti-Inflammatory Effects

**DOI:** 10.3390/ijms26083561

**Published:** 2025-04-10

**Authors:** Anqi Zeng, Yunyun Quan, Hongxia Tao, Ying Dai, Linjiang Song, Junning Zhao

**Affiliations:** 1Translational Chinese Medicine Key Laboratory of Sichuan Province, Sichuan Institute for Translational Chinese Medicine, Sichuan Academy of Chinese Medicine Sciences, Chengdu 610041, China; zeng6002aq@163.com (A.Z.); quanyunyun1118@163.com (Y.Q.); daiying_77@126.com (Y.D.); 2Sichuan Institute for Translational Chinese Medicine, Chengdu 610041, China; 3West China Hospital, Sichuan University, Chengdu 610041, China; taohongxia1220@163.com; 4School of Medical and Life Sciences, Chengdu University of Traditional Chinese Medicine, Chengdu 611137, China

**Keywords:** tetrahydrocurcumin, inflammation, tumor, neurodegenerative disease

## Abstract

Tetrahydrocurcumin (THC), a curcumin derivative, shows potential in oncology and neurology. It regulates NF-κB, reduces inflammation, promotes cancer cell apoptosis, inhibits tumor angiogenesis, and enhances antioxidants, aiding in treating inflammation-related cancers. In neurology, THC’s anti-inflammatory and antioxidant properties protect neurons, reduce neuroinflammation, and support autophagy for cellular debris clearance, with its blood–brain barrier penetration offering a neuroprotective edge. Research on THC’s therapeutic application must focus on improving delivery and bioavailability and confirming its clinical safety and efficacy.

## 1. Introduction

Tetrahydrocurcumin (THC), a principal metabolite of curcumin (CUR), exhibits superior bioavailability and a spectrum of pharmacological activities that surpass those of its parent compound [1]. The principal metabolic pathways of THC within the organism encompass antioxidant activity, hypoglycemic and hypolipidemic effects, along with other therapeutic functions. Furthermore, THC demonstrates neuroprotective potential, exemplified by its capacity to modulate the Ras/ERK signaling pathway, thereby alleviating cell cycle arrest and inhibiting apoptosis of microglia in Alzheimer’s disease models [2,3,4,5]. Regarding the bioavailability of THC, studies indicate that its solid dispersion formulation significantly enhances bioavailability in mice compared to the compound in its isolated form [6,7]. This enhancement is attributed to a modest extension in mean residence time (MRT) and elimination half-life (t1/2), coupled with a marked reduction in the time to reach peak concentration (tmax), thereby augmenting plasma concentrations of THC [8]. This finding holds profound implications for elevating the clinical utility of THC, as it may amplify its efficacy while minimizing the requisite dosage. However, certain studies have highlighted that THC may exhibit a “hormesis effect”, wherein low doses may foster the proliferation of tumor cells, while higher doses can suppress tumor growth and modulate the tumor microenvironment. As a metabolite of curcumin, THC retains the diverse biological activities of its precursor and further potentiates its pharmacological effects by enhancing bioavailability [9,10,11]. Despite potential adverse effects, the therapeutic potential of THC can be optimized through precise dosage control and the application of suitable administration modalities. Future research endeavors should delve deeper into the safety and efficacy of THC to harness the full potential of this promising therapeutic agent.

Inflammation assumes a multifaceted role in the genesis of tumors and neurodegenerative diseases, implicating cytokines, immune cells, and the intricate network that interacts with these cellular and molecular elements [12,13,14]. Persistent inflammatory responses, stemming from chronic inflammation or tissue injury, foster cellular transformation via genetic damage or pro-inflammatory agents, thereby precipitating chronic inflammation and tumorigenesis [15,16]. Tumor-extrinsic inflammation arises from a variety of factors, including bacterial and viral infections, autoimmune disorders, obesity, smoking, asbestos exposure, and excessive alcohol consumption, all of which elevate the risk of cancer and spur malignant progression [17,18,19,20]. Inflammation is not only correlated with the emergence of tumors but also exerts a significant influence on their progression. For instance, inflammatory cells within the tumor microenvironment, such as macrophages and microglia, promote the proliferation, invasion, and metastasis of tumor cells by secreting inflammatory mediators like interleukins and tumor necrosis factors [21,22,23,24,25]. Furthermore, inflammation impacts immune surveillance and responsiveness to therapy. Tumors associated with inflammation typically exhibit enhanced angiogenic potential, facilitating the survival and dissemination of tumor cells. For example, mast cells promote angiogenesis in squamous epithelial carcinogenesis by activating specific enzymes. Moreover, inflammation can also foster drug resistance in tumor cells by altering the tumor microenvironment [26,27,28,29].

In the context of neurodegenerative afflictions, pivotal inflammatory mediators, including IL-1β and IL-18, are instrumental in the subsequent release post-neural injury [30]. Such inflammatory mediators are capable of instigating an immune response within the nervous system, culminating in neuronal demise and impaired functionality [31]. The NLRP3 inflammasome constitutes a vital element of the innate immune apparatus, with its activation precipitating neuroinflammation—a critical factor in the etiology and advancement of neurodegenerative pathologies such as Alzheimer’s, Parkinson’s, amyotrophic lateral sclerosis, and multiple sclerosis [32,33]. Empirical research indicates that persistent infections of bacterial, viral, and fungal origins may be contributory risk factors for the onset of neurodegenerative diseases. Such infections are capable of inducing chronic neuroinflammation, which can precipitate metabolic disturbances and a spectrum of other pathological conditions [34,35,36]. Furthermore, inflammation can intensify the trajectory of neurodegenerative diseases by modulating the immune response and the mechanisms governing cell death within the nervous system.

To encapsulate, inflammation exerts a complex and pivotal influence on the evolution of both oncological and neurodegenerative conditions. It partakes not solely in the incipient phases of these pathologies but also significantly contributes to their progression and exacerbation [37,38]. Consequently, therapeutic interventions that target inflammatory mechanisms, including the employment of anti-inflammatory pharmaceuticals or alternative modalities, may offer innovative strategies for the prophylaxis and amelioration of such afflictions. As a burgeoning anti-inflammatory and neuroprotective entity, the prospective applications of THC in the mitigation of inflammation-driven oncogenesis and neurodegenerative disorders warrant additional investigation. This treatise elucidates the role of THC in addressing inflammation-induced carcinogenesis and neurodegeneration, with the ambition of furnishing direction for the more efficacious development and application of this compound.

## 2. Chemical and Biological Properties of THC

### 2.1. Chemical Structure of THC Compared with Curcumin

THC serves as the principal metabolite of curcumin (CUR) in the human body, exhibiting superior antioxidant efficacy and bioavailability through gastrointestinal absorption compared to its precursor. Curcumin, a lipophilic bioactive polyphenolic compound with the molecular formula C_21_H_20_O_6_ (depicted in Figure 1), is derived from the rhizomes of Curcuma longa, a revered medicinal plant in traditional Chinese medicine. In contrast to curcumin, the chemical architecture of THC arises from the saturation of methylene bridges within the β-diketone core of curcumin through hydrogenation, yielding the THC molecule [39,40].

The synthetic routes to THC commence with vanillin and acetone, employing Claisen condensation to synthesize curcumin, which is subsequently subjected to a palladium-catalyzed hydrogenation process to yield THC. Furthermore, microbial transformation techniques are harnessed to alter the molecular scaffold of curcumin, efficiently transforming it into an array of modified derivatives, among which THC is notably included [41].

In contrast to curcumin, THC demonstrates enhanced pharmacological potency, particularly in its antioxidant, hypoglycemic, and hypolipidemic capabilities [39,42]. This enhanced activity can be attributed to the improved stability and bioavailability of THC within the physiological milieu, thereby conferring more robust efficacy in these domains [11]. Furthermore, THC manifests neuroprotective properties, adept at mitigating G1/S cell cycle arrest in Alzheimer’s disease paradigms and modulating microglial cell cycling. It also downregulates the expression of tumor necrosis factor α, upregulates TGF-β1 levels, and synergistically impedes both cell cycle stagnation and apoptosis [43,44,45]. To encapsulate, as the principal metabolite of curcumin, THC maintains the diverse biological activities of its precursor while demonstrating superior performance in specific aspects, including its heightened antioxidant potential and enhanced bioavailability. These attributes render THC an asset in the realms of medical and health product development.

### 2.2. Solubility, Stability, and Bioavailability of THC

As the principal metabolite of curcumin within the body, THC possesses more robust antioxidant activity and gastrointestinal absorption than curcumin, along with an array of biological activities that surpass those of its precursor. Nonetheless, the solubility, stability, and bioavailability of curcumin and its derivatives, including THC, represent pivotal factors that constrain their clinical utility [39]. Curcumin, a naturally occurring pigment characterized by low solubility and instability, can undergo enhancement of its aqueous solubility and stability through techniques such as solid dispersion and microencapsulation [46]. The bioavailability of THC can be augmented via a spectrum of pharmaceutical delivery systems, encompassing nanoparticles, liposomes, micelles, self-microemulsions, solid dispersions, and phospholipids. Empirical studies have demonstrated that solid dispersions of THC can markedly enhance the bioavailability in mice when administered via gavage, as opposed to the administration of THC in its unadulterated form. Furthermore, refining the biotransformation process to optimize THC yield can substantially augment production, thereby indirectly enhancing bioavailability. The solubility, stability, and bioavailability of THC can be ameliorated through a multitude of approaches. Such enhancements not only bolster its in vivo efficacy but also broaden its applicability within the domains of food and medicinal sciences. Future investigative endeavors should delve into diverse solubilization and stabilization methodologies to harness its utmost potential for clinical deployment [11,47].

### 2.3. Anti-Inflammatory Mechanisms of THC

#### 2.3.1. Inhibiting Inflammatory Factors and Signaling Pathways

THC potently curbs the genesis of a spectrum of inflammatory cytokines, primarily through the modulation of the expression of a plethora of inflammatory mediators and signaling cascades, including TNF-α, IL-6, and MIP-2. Furthermore, THC is capable of attenuating inflammatory responses via a multitude of distinct mechanisms [11,48,49]. Additionally, THC efficaciously mitigates the excessive secretion of MCP-1 and IL-6 in vascular smooth muscle cells incited by lipopolysaccharide (LPS), in addition to dampening the heightened expression of TLR4 and decreasing the phosphorylation levels within the MAPKs signaling pathway [50]. Collectively, these observations suggest that THC modulates the expression of inflammatory factors by impacting particular signaling pathways, namely TLR4, MAPKs, and NF-κB. Moreover, THC can diminish inflammation by impeding the activation of the JAK/STAT, Nrf2/HO-1, and JNK/ERK signaling pathways [50,51,52]. Within specific experimental paradigms, THC has also manifested inhibitory actions on the NF-κB/VEGF/MMP-9 signaling axis, thus ameliorating cerebral edema and inflammation induced by acute hypobaric hypoxia [53]. These revelations lay a theoretical foundation for the continued exploration of the therapeutic application of THC in the context of inflammation-associated diseases (Figure 2).

#### 2.3.2. Improved Vascular Function and Structure

THC ameliorates hypertension, arteriosclerosis, and vascular remodeling induced by cadmium, conditions frequently linked to a proliferation of smooth muscle cells, excessive collagen deposition, and heightened matrix metalloproteinase (MMP)-2 and MMP-9 levels. THC exerts a protective effect by augmenting the bioavailability of nitric oxide (NO), mitigating oxidative stress, and facilitating the amelioration of vascular remodeling processes [54,55].

#### 2.3.3. Antioxidant Effect

When inflammation occurs, the production of free radicals such as reactive oxygen species (ROS) and active nitrogen (RNS) in the body increases, and the antioxidant defense system may not be able to effectively neutralize these free radicals, leading to oxidative stress. Oxidative stress damages cellular components such as DNA, proteins, and lipids, which in turn triggers and maintains an inflammatory response [56]. Antioxidants are able to neutralize free radicals and reduce oxidative stress, thereby regulating the inflammatory response to some extent. Antioxidants such as vitamin C, vitamin E, beta-carotene, and selenium, for example, can reduce the damage to cells caused by oxidative stress, which in turn relives inflammatory symptoms [56].

THC functions as a robust antioxidant, curtailing the generation of reactive oxygen species (ROS) and mitigating the oxidative stress and fibrosis engendered by elevated glucose levels through the activation of the SIRT1 signaling pathway. Furthermore, THC augments the capacity to withstand hypoxia by bolstering superoxide dismutase (SOD) activity [42,53,57].

#### 2.3.4. Modulating Immune Response

THC modulates allergic responses in asthmatic mice through the regulation of the gut microbiome. Moreover, THC diminishes the synthesis of platelet thromboxane A2 (TxA2) by impeding the MAPKs/cPLA2 signaling pathway, consequently mitigating the occurrence of thrombosis [58].

### 2.4. Potential of THC in Cancer Prevention and Therapy

#### 2.4.1. Mechanism of Action of THC on Cancer Cells

##### Anti-Proliferative Effect

THC possesses the capacity to inhibit the proliferation of tumor cells. Studies have demonstrated that THC exerts a discernible inhibitory effect on breast cancer cells MCF-7; its impact on the proliferative capacity of MCF-7 cells, as ascertained by CCK8 assays and clonal formation tests, suggests a potential anti-proliferative action. Furthermore, investigations into CT26 colorectal cancer cells have illustrated that THC can markedly suppress both the quantity and dimensions of tumor cell clonal formations in a concentration-dependent fashion [59,60].

##### Proapoptotic Effect

THC facilitates the apoptosis of tumor cells via multiple mechanisms. A study has reported that THC can diminish the expression of Bcl-2/Bax proteins, thereby triggering the apoptosis of tumor cells. Subsequent research indicated that treatment with THC induced morphological alterations in CT26 colorectal cancer cells, culminating in cellular apoptosis. This treatment also notably increased the expression of BAX and activated caspase 3, while decreasing the expression of BCL-2 protein. These findings imply that THC advances the apoptotic process in tumor cells by modulating the expression of pivotal proteins associated with apoptosis [9,59].

##### Anti-Angiogenic Effect

THC also exerts anti-angiogenic effects, playing a pivotal role in the suppression of tumor growth and metastasis. Bhornprom Yoysungnoe et al. have demonstrated that the combinatorial treatment with THC and celecoxib leads to the inhibition of tumor growth and angiogenesis through the downregulation of VEGF, COX-2, and EGFR expression. Nonetheless, this combined therapeutic approach did not manifest a synergistic effect on curbing tumor growth and angiogenesis within the cervical cancer (CaSki)-implanted nude mice model [61]. Pornprom Yoysungnoen et al. investigated the anti-cancer and anti-angiogenic properties of THC in a model of implanted hepatocellular carcinoma in nude mice. Their findings indicate the presence of pathological angiogenic characteristics, such as microvascular dilation, tortuosity, and hyperpermeability. Both curcumin (CUR) and THC were found to mitigate these pathological features. In the HepG2 groups, the chorioallantoic membrane (CAM) vascularity index (CV) was significantly elevated on days 7 (52.43%), 14 (69.17%), and 21 (74.08%) compared to the control group (33.04%, *p* < 0.001). Administration of CUR and THC led to a substantial reduction in the CV, with respective *p*-values of <0.005 and <0.001. Notably, the anti-angiogenic effects of both CUR and THC exhibited a dose-dependent response. Nonetheless, a more pronounced beneficial effect of THC over CUR treatment was discernible, particularly in the CV measurements on day 21 (44.96% and 52.86%, respectively, *p* < 0.0575) [62] (Figure 3 and Table 1).

#### 2.4.2. Research Progress of THC in Different Cancer Models

##### Cervical Cancer

Cervical cancer represents the most prevalent malignant gynecological tumor. Investigations have revealed that THC possesses the capacity to inhibit tumor angiogenesis [63,64]. In vivo experimentation involved administering oral doses of 100, 300, and 500 mg/kg of THC to mice daily for 30 days. The findings indicated that THC notably suppressed tumor angiogenesis, as well as tumor volume and growth rate, in the treatment cohort. This inhibitory effect is likely mediated through the downregulation of Hypoxia-Inducible Factor-1 alpha (HIF-1-α).

##### Breast Cancer

Breast cancer, a malignancy arising in the epithelial tissues of the breast, is witnessing a rising incidence and affecting increasingly younger demographics annually. Breast cancer cells readily diminish in their intercellular adhesion capabilities, proliferate and disseminate with ease, swiftly metastasize to organs including bone marrow and lymph nodes, and ultimately pose a significant threat to patients’ lives [65]. Researchers discovered that THC exerted a discernible inhibitory effect on the in vitro proliferation of MCF-7 cells, and that THC administration could facilitate the apoptosis of these cells and diminish their metastatic potential. The underlying mechanism is likely the downregulation of Bcl-2/Bax protein expression to trigger tumor cell apoptosis, coupled with the suppression of matrix metalloproteinases MMP-2 and MMP-9 expression to impede cell metastasis [66,67]. Concurrently, Ning Kang et al. also determined that THC significantly restrained cell growth by instigating mitochondrial-mediated apoptosis and inducing G2/M cell cycle arrest in MCF-7 cells. Concurrently, they also determined that THC significantly restrained cell growth by instigating mitochondrial-mediated apoptosis and inducing G2/M cell cycle arrest in MCF-7 cells [67].

##### Liver Cancer

Hepatocellular carcinoma (HCC) poses a significant global health challenge, characterized by a high incidence and propensity for progression. It is also the predominant cause of mortality among individuals with cirrhosis. The researchers have demonstrated that THC markedly enhances the survival rate of mice bearing hepatocellular carcinoma in in vivo assays, suppresses the proliferation of HCC cells, and diminishes the ascites volume and abdominal girth associated with liver cancer [68]. Mechanistic studies have revealed that THC significantly activates and induces the cleavage of caspase-3 and caspase-9, elevates the expression of the p53 gene, facilitates the apoptosis of HT22 liver cancer cells, and exerts antitumor effects. Additionally, Pornprom Yoysungnoen et al. discovered that THC impedes the proliferation of liver cancer cells by exerting anti-angiogenic properties [62].

##### Acute Myeloid Leukemia

Acute leukemia represents the predominant form of cancer among children, comprising 26% of total cancer diagnoses and 20% of cancer-related fatalities in the pediatric population. The majority of anti-cancer agents are designed to induce cancer cell death, yet, the emergence of multidrug resistance frequently culminates in chemotherapy failure. Studies have revealed that THC primarily elicits the demise of drug-resistant HL60 cells via the induction of apoptosis and autophagy [69]. Concurrently, THC triggers autophagic cell death in HL-60 promyelocytic leukemia cells by augmenting the development of acidic vascular organelles (AVOs), which serve as markers of autophagy [70]. These findings suggest that THC harbors potential therapeutic value in the treatment of acute myeloid leukemia.

##### Fibrosarcoma

Fibrosarcoma is a malignant tumor that originates from fibroblasts and is characterized by the production of collagen fibers. It is commonly observed in young adults, exhibiting rapid growth and a propensity for hematogenous metastasis. Research has demonstrated that an escalation in THC treatment concentration correlates with a significant diminution in the invasive and migratory capabilities of HT1080 human fibrosarcoma cells [71]. Additionally, THC diminished cellular adhesion to both matrix and lamin-coated surfaces. Enzymatic assessments revealed that THC treatment led to a reduction in the levels of matrix metalloproteinases 2 and 9 (MMP-2, MMP-9), as well as urokinase plasminogen activator (uPA). THC exerts an inhibitory effect on the expression of matrix metalloproteinase1 (MT1-MMP) and MMPs and is a tissue inhibitor of metalloproteinases 2 (TIMP-2) proteins. The findings collectively indicate that THC can markedly suppress the metastatic potential of fibrosarcoma cells in vitro.

##### Colon Cancer

Colon cancer is a prevalent malignant neoplasm of the digestive tract, specifically arising in the colon, and is the third most frequently occurring gastrointestinal tumor. Research has determined that the in vitro IC_50_ value of THC for human colorectal adenocarcinoma cells (HT-29) is 28.67 ± 1.01 μg/mL, and the inhibitory selectivity index of THC for HT-29 cells is fourfold higher compared to that for normal colorectal epithelial cells [72]. Concurrently, THC demonstrates a significant chemopreventive efficacy against azoxymethane (AOM)-induced colon carcinogenesis in murine models [60]. Collectively, these findings substantiate that THC exerts a favorable chemopreventive influence on the genesis of colon tumors.

##### Osteosarcoma

Osteosarcoma represents the most prevalent primary malignant bone tumor among children and adolescents. Nonetheless, the absence of early biomarkers renders it challenging to detect, resulting in the majority of patients being diagnosed with high-grade malignancies. Human osteosarcoma is classified as a malignant tumor with a poor prognosis and a propensity for metastasis. Research has identified that THC can markedly diminish the proliferation of osteosarcoma cells and curb their migration and invasion within a nude mouse model of lung metastasis [73]. Furthermore, THC fosters the mesenchymal-to-epithelial transition (MET) process. Studies have illuminated that hypoxia-inducible factor-1 alpha (HIF-1α) is pivotal in the anti-metastatic effects of THC. Notably, THC downregulates HIF-1α expression by impeding the Akt/mTOR and p38 MAPK signaling pathways. Moreover, THC substantially suppressed HIF-1α expression and angiogenesis under hypoxic conditions. THC activates autophagy, induces MET, and curbs angiogenesis, mechanisms that are interrelated with HIF-1α regulation.

##### Non-Small Cell Lung Cancer

Lung cancer is the predominant cause of cancer-related mortality globally. The most prevalent variant of lung cancer is non-small cell lung cancer (NSCLC). Presently, therapeutic options for non-small cell lung cancer are rather constrained, encompassing chemotherapy, radiation therapy, surgical resection, and targeted therapies. The study has discovered that THC can suppress the growth and proliferation of A549 cells. THC treatment significantly augments autophagy [74]. Reverse transcription quantitative polymerase chain reaction (RT-qPCR) analysis indicated an elevation in Beclin-1 expression following THC treatment. The ratio of microtubule-associated protein 1A/1B-light chain 3 (LC3)-II to LC3-I was diminished. Cellular protein expression levels of diverse autophagy markers, including p62, phosphorylated mammalian target of rapamycin (p-mTOR), phosphatidylinositol-3 kinase (PI3K), phosphorylated PI3K (P-PI3K), protein kinase B (Akt), and phosphorylated Akt (P-Akt), were substantially decreased. These findings suggest that THC induces autophagy by impeding the PI3K/Akt/mTOR signaling pathway, thereby suppressing the progression of NSCLC.

##### Glioma

Glioma is among the most prevalent and malignant brain tumors within the central nervous system. Currently, radiotherapy remains the most efficacious treatment modality for glioma. Regrettably, the low radiosensitivity of tumor cells often leads to unsatisfactory radiotherapy outcomes, with gliomas having dismal relative survival rates [76]. Utilization of radiosensitizers can elevate the rate of glioma cell death, thereby enhancing the therapeutic efficacy of radiotherapy. Studies have revealed that, in comparison to the radiotherapy-only group, there was a marked reduction in tumor cell viability and an increase in the rate of apoptosis within the group administered THC in conjunction with radiotherapy. Cells treated with combined THC and radiation demonstrated lower cell viability and a higher apoptosis rate compared to the radiation group. Moreover, the intracellular GSH was also decreased in the THC co-treated C6 cells. More importantly, the combinatorial treatment group significantly induced G0/G1 cell cycle arrest and a decrease in the S phase cell through the downregulation of cyclin D1 and PCNA. The in vivo therapeutic efficacy assay indicated that the growth of the tumor was greatly inhibited in the combinatorial group [76]. Furthermore, the intracellular glutathione (GSH) levels in C6 cells treated with THC were also observed to diminish. Most importantly, the combination therapy group significantly induced G0/G1 cell cycle arrest and a reduction in the S-phase cell population through the downregulation of Cyclin D1 and proliferating cell nuclear antigen (PCNA) expression. These findings suggest that THC can act synergistically to augment the radiosensitivity of glioma cells.

### 2.5. Protective Effects of THC in Neurodegenerative Diseases

#### 2.5.1. Potential Mechanisms of THC on Neuroprotection and Repair

##### Antioxidation

THC expedites the clearance of free radicals within the body by activating a suite of antioxidant enzymes, including superoxide dismutase (SOD), glutathione peroxidase (GSH-Px), catalase (CAT), and glutathione S-transferase (GST), thereby reinforcing the stability of the brain’s antioxidant enzyme system [5]. Furthermore, THC diminishes the concentration of reactive oxygen species (ROS) and shields hippocampal cells against oxidative injury. These salutary effects contribute to the amelioration of neuronal damage instigated by free radicals and enhance the viability of neuronal cells [77,78,79].

##### Anti-Inflammatory Effect

THC can modulate the TLR-4/P38/MAPK signaling pathway and attenuate the levels of IL-6, TNF-α, and other inflammatory mediators, thereby dampening the inflammatory response. Moreover, THC can downregulate the expression of tumor necrosis factor α (TNF-α) and upregulate the expression of transforming growth factor β1 (TGF-β1), synergistically inhibiting cell cycle arrest and apoptosis. These multifaceted effects contribute to the enhancement of neuronal function and the reduction in neuronal mortality [49,50].

##### Inhibition of Apoptosis

THC can diminish the activity of caspase 3 by activating the mitogen-activated protein kinase (MAPK) signaling pathway, suppress the expression of pro-apoptotic proteins, and ultimately curb apoptosis triggered by cerebral ischemia–reperfusion injury. Furthermore, THC can ameliorate autophagy by reducing the release of cytochrome c and homocysteinylation, thereby conferring neuroprotective effects [70,79].

##### Other Potential Mechanisms

THC may also modulate the PI3K/Akt/mTOR signal transduction pathway, curb the accumulation of edema in the spinal cord, and mitigate inflammatory mediators, thereby effectively downregulating the gene expression of matrix metalloproteinases 3 and 13 and cyclooxygenase-2 (COX-2), facilitating the phosphorylation of Akt, augmenting the expression of forkhead box (FOX) O4 protein, and thereby achieving spinal cord protection [57]. In summary, THC can efficaciously shield neuronal cells from a spectrum of injurious stimuli through its anti-inflammatory and antioxidant properties. This neuroprotection is mediated by the activation of antioxidant enzyme systems, attenuation of inflammatory mediator levels, suppression of apoptotic pathways, enhancement of autophagy, and other related mechanisms (Figure 4 and Table 2).

#### 2.5.2. Potential Use of THC in Neurodegenerative Diseases

##### Brain Injury

Traumatic brain injury (TBI) represents a significant health issue, characterized by high rates of morbidity and mortality. The neuroprotective effects of THC in experimental models of traumatic brain injury (TBI) have been established, primarily through the inhibition of oxidative stress, mitigation of mitochondrial dysfunction, and preservation against cellular apoptosis [80]. In 2018, Guan Wei et al. [81] investigated the neuroprotective efficacy of THC in rats subjected to traumatic brain injury, along with its underlying mechanisms. Post-THC administration, the researchers assessed neurological scores, cerebral water content, and neuronal degeneration in the cerebral cortex. Cerebral tissue samples were harvested subsequent to the neurological assessments for further analytical examination. The findings indicated that THC treatment could mitigate cerebral edema, reduce TBI-induced neuronal apoptosis, and enhance neurobehavioral performance. Following THC treatment, the expression of Nrf2 was found to be upregulated in response to TBI. These findings propose that THC may ameliorate neuronal function following traumatic brain injury through the activation of the Nrf2 signaling pathway.

##### Cerebral Edema

High-altitude cerebral edema (HACE), frequently regarded as an advanced manifestation of acute mountain sickness (AMS), affects individuals subjected to acute hypobaric hypoxia (AHH) in the absence of acclimatization [82]. HACE constitutes a grave and potentially lethal high-altitude affliction. Yang Pan et al. [53] determined that the prophylactic administration of THC (40 mg/kg) over a three-day period could markedly reduce the cerebral water content (BWC) induced by acute hypobaric hypoxia (AHH) and diminish the concentrations of interleukin-1β (IL-1β) and tumor necrosis factor α (TNF-α). Elevate the levels of superoxide dismutase (SOD) and augment the body’s hypoxia resistance. Histological and ultrastructural assessments of brain tissue revealed that THC mitigated AHH-induced pericellular edema and diminished perivascular space, thereby concurrently reducing edema and safeguarding brain mitochondria. In vitro experimentation indicated an upregulation of IL-1β expression 24 h post-implantation, succeeded by an escalation in vascular endothelial growth factor (VEGF) levels. Furthermore, THC substantially downregulated the expression of VEGF, matrix metallopeptidase-9 (MMP-9), and nuclear factor-kappa B (NF-κB) in astrocytes subjected to hypoxic conditions (4% O2). Collectively, these findings imply that THC possesses potential preventative efficacy against HACE, which is associated with the suppression of the NF-κB/VEGF/MMP-9 signaling pathway.

##### Cerebral Ischemia

Cerebral ischemia is a pathological phenomenon that induces damage to localized brain tissue and its functional integrity. The severity of the damage correlates with the duration of ischemia and the extent of residual blood flow. Short-term, incomplete ischemia typically results in reversible injury, whereas prolonged, complete, or severe ischemia can lead to infarction. Cerebral ischemia followed by reperfusion can lead to significant impairment of brain function. During cerebral ischemia, alterations in the bioelectric properties of neuronal cells occur, accompanied by the emergence of pathological slow waves. Upon reperfusion after a period of ischemia, these slow waves persist and intensify [83]. In a study conducted by Nandan K. Mondal [84], eight 10-week-old male C57BL/6 wild-type mice received a daily intraperitoneal injection of THC (25 mg/kg) for three consecutive days, commencing after a 40 min reperfusion period following middle cerebral artery occlusion (MCAO), and continued for 72 h thereafter. Post-THC treatment, significant enhancements in cerebral function and motor coordination were observed in the ischemic mice, with reductions in neurological deficit scores, infarct volume, cerebral edema, and microvascular leakage in the parenchyma. Following treatment, there were significant alterations in total homocysteine (tHcy) levels, homocysteine metabolic enzymes, and mitochondrial oxidative stress. Matrix metalloproteinase-9 (MMP-9) activity was elevated, and the expression of tight junction proteins was reduced. Additionally, notable variations in autophagy markers, as well as proteins involved in fusion and fission events, were identified. Experimental evidence has confirmed that THC ameliorates mitochondrial dysfunction in cerebral vasculature during ischemic stroke via epigenetic mechanisms [85]. The experimental findings of Neetu Tyagi et al. also demonstrated that THC exerts neuroprotection against homocysteine-induced neurotoxicity and enhances autophagy in ischemia/reperfusion injury by mitigating the homocysteinylation of cellular components under hyperhomocysteinemic pathological conditions [85]. Bin Lin et al. also discovered that THC attenuated the activation of the ERK signaling pathway triggered by ischemia/reperfusion (I/R) injury and diminished the phosphorylation of GRASP65. This can induce changes in ERK and GRASP65 phosphorylation levels. THC demonstrated a protective effect on cerebral ischemia/reperfusion injury [86].

##### Parkinson’s Disease

Parkinson’s disease (PD) is a prevalent neurodegenerative disorder with a higher incidence in the elderly, typically manifesting around the age of 60 years. In China, the prevalence of PD among individuals over the age of 65 is approximately 1.7%. The principal pathological hallmark of Parkinson’s disease is the degeneration and demise of dopaminergic neurons within the substantia nigra of the midbrain. This results in a marked decrease in dopamine (DA) levels in the striatum, thereby precipitating the disease [87]. In a study utilizing the 1-methyl-4-phenyl-1,2,3,6-tetrahydropyridine (MPTP)-induced mouse model of Parkinson’s disease, A. Radjeswari et al. observed that the administration of THC (60 mg/kg) could notably counteract the MPTP-induced depletion of dopamine (DA) and dihydroxyphenylacetic acid (DOPAC). The findings indicated that THC exerts neuroprotective properties against MPTP-induced neurotoxicity and confers a measurable ameliorative effect on Parkinson’s disease [88].

##### Alzheimer’s Disease

Improve the Neurotoxicity Caused by Aβ

Alzheimer’s disease (AD) is a progressively debilitating neurodegenerative disorder with a subtle onset. AD is a neurodegenerative condition influenced by a complex interplay of genetic and non-genetic factors. AD is typified by the presence of amyloid-beta (Aβ) plaques and neurofibrillary tangles. The hallmark pathologies of AD include the substantial deposition of amyloid-beta (Aβ) plaques, the formation of tau protein neurofibrillary tangles, and neuroinflammation. These features are widely acknowledged and are ultimately responsible for synaptic dysfunction, neuronal demise, and the cognitive decline observed in AD patients [89]. Shilpa Mishra et al. discovered that THC can mitigate the Aβ-induced escalation in reactive oxygen species, attenuate the reduction in mitochondrial membrane potential, and inhibit caspase activation [90]. THC also confers protection to human neurons against Aβ-induced toxicity. THC exerts a protective effect against Aβ-induced neurotoxicity. Yu Xiao et al. further corroborated that THC can ameliorate the apoptosis of glial cells provoked by Aβ in in vitro experiments. In in vivo experiments, THC was shown to enhance the learning and memory capabilities of APP/PS1 transgenic mice and to diminish the hippocampal Aβ content. Proteomic analysis of the mouse hippocampus indicates that the impact of THC on neuronal apoptosis is predominantly associated with the “Ras signaling pathway”. These studies offer novel insights and research directions regarding the potential of THC to mitigate the progression of AD [91].

Neuroprotective Effect

Elevated levels of homocysteine (Hcy), termed hyperhomocysteinemia (HHcy), represent a significant risk factor for a multitude of neurological disorders. HHcy is also recognized as a contributing risk factor for neurodegenerative diseases [92]. This is attributed to the susceptibility of the mercaptan (SH) group present in Hcy to oxidation, thereby generating reactive oxygen species (ROS) that precipitate oxidative stress. A hallmark of numerous neurodegenerative diseases is the presence of mitochondrial dysfunction and localized neuronal cell death within the nervous system [76]. Mitochondria play an essential role in neuronal survival and function, with their dysfunction being a pivotal factor in the pathogenesis of neurological diseases. Mitochondria, as morphologically dynamic organelles, fulfill diverse roles in cellular processes related to survival and demise, including metabolite synthesis, apoptosis, and energy production [93]. Jonathan C. et al. discovered that THC could ameliorate mitochondrial remodeling in rat brain endothelial cells (bEnd3) induced by homocysteine (Hcy). Moreover, bEnd3 cells were subjected to Hcy treatment, both with and without the presence of THC [93]. The pretreatment of bEnd3 cells with THC (15 μM) enhanced the cells’ resilience to Hcy-induced oxidative stress, mitochondrial dynamics, and the process of mitophagy. Chang-Hyun Park et al. determined that THC also exerted a protective effect on HT22 hippocampal neuronal cells in response to glutamate excitotoxicity, mitigating HT22 cell death and demonstrating robust antioxidant properties [94]. Furthermore, THC significantly attenuated the intracellular calcium ion influx heightened by glutamate. Additionally, THC markedly decreased the intracellular buildup of oxidative stress instigated by glutamate. Moreover, THC substantially inhibited the apoptotic processes in HT22 cells. The experimental outcomes indicate that THC is a potent neuroprotective agent, safeguarding neuronal cells from glutamate-induced injury.

### 2.6. Clinical Applications and Safety Considerations

#### Dose, Mode of Administration, and Potential Side Effects of THC

As the principal metabolite of curcumin within biological systems, THC exhibits a plethora of biological activities, encompassing antioxidant, neuroprotective, oncostatic, hypoglycemic, and lipid-lowering effects. The dosing regimen, mode of administration, and potential side effects of THC constitute a focal point of investigative efforts. The therapeutic spectrum of THC dosages is notably broad. Varied dosing paradigms have been employed across studies, spanning a low to high dosage continuum. Illustratively, within a study aimed at mitigating doxorubicin-induced cardiotoxicity, a dosage of 100 mg/kg was administered daily for a quintuple period via gavage. Within the purview of research investigating cognitive and mnemonic deficits engendered by the amalgamation of D-galactose and aluminum trichloride in murine models, different groups were given different concentrations of drugs (20 mg/kg, 60 mg/kg, and 200 mg/kg) during a 35-day basis. This infers that the efficacious dosage of THC may be contingent upon the objectives and the experimental paradigm under scrutiny. The modalities for THC administration are predominantly gavage and intraperitoneal injection. The intragastric route is frequently employed for drug delivery, exemplified in a study involving db/db mice exhibiting spontaneous type 2 diabetes, wherein micro-powdered THC was intragastrically administered over a six-week duration. In the realm of dermatological research focusing on the reparative effects on photoaged murine skin, daily gavage was similarly utilized. Furthermore, intraperitoneal injection represents another prevalent administration method, as evidenced in breast cancer research where varying concentrations of THC were intraperitoneally dispensed [59]. While THC may bestow a multitude of potential health benefits, vigilance regarding its possible side effects is imperative and cannot be overlooked. Within the context of breast cancer research, it was observed that the tumor volume in the low-dose cohort of tumor-bearing mice exhibited an increment relative to the control group. Conversely, in the high-dose cohort, a reduction in tumor volume post-administration was noted, indicating a dose-dependent therapeutic response and underscoring the inherent risk of side effects [59]. Moreover, curcumin and its derivatives are characterized by their exceedingly low aqueous solubility and pronounced lipophilicity, which collectively restrict their oral bioavailability, culminating in suboptimal gastrointestinal absorption. Consequently, despite the favorable biological activity of THC, prudence in dose regulation and vigilance for potential side effects are essential in the clinical realm [95,96,97]. The dosage, mode of administration, and potential side effects of THC exhibit a spectrum of variability, necessitating the delineation of the most apt utilization strategy predicated upon the specific scientific aims and experimental models. Concurrently, a rigorous assessment of its potential side effects is imperative to guarantee its safety and efficacy within the clinical therapeutic spectrum.

### 2.7. Drug Delivery Systems and Pharmacokinetics

#### Nanoparticle

Vandita Kakkar and associates have delved into the anti-inflammatory potential of a THC gel by encapsulating THC within a nanocarrier system, ultimately administering it in the form of a hydrogel. The THC lipid nanoparticles (THC-SLNs), fabricated through microemulsification, assumes an elliptical morphology, as evidenced by transmission electron microscopy, with a mean particle size of 96.6 nm and a zeta potential of −22 millivolts. The THC-SLNs exhibited a drug content of 94.51% ± 2.15% and an encapsulation efficiency of 69.56% ± 1.35%. Thermal analysis via differential scanning calorimetry and X-ray diffraction confirmed the successful formation of THC-SLNs. In vitro drug release kinetics revealed that the THC-SLNs gels adhered to the Higuchi model, indicative of non-Fickian diffusion. Transdermal permeation studies demonstrated that the THC-SLNs gels facilitated approximately 17-fold enhanced skin penetration compared to gels containing free THC. Assessments of skin irritation, occlusivity, and stability indicated that the formulation is non-irritating, stable, and possesses desirable occlusive attributes. Pharmacodynamic evaluation utilizing an excised wound mouse model categorically illustrated the augmented anti-inflammatory efficacy of the THC-SLN gel, a finding corroborated by biochemical and histopathological analyses. Remarkably, the THC-SLN gels exhibited significantly superior activity (*p* ≤ 0.001) when contrasted with gels containing unencapsulated THC. Given that inflammation is a fundamental component of myriad skin pathologies, this innovative product development heralds novel therapeutic pathways for the treatment of a spectrum of dermatological disorders [98].

### 2.8. Future Research Directions and Challenges

#### 2.8.1. Development of New Dosage Forms

In recognition of the pharmacological potency of THC, as well as its inherent challenges such as tenuous stability and limited bioavailability, there has been an impetus to develop advanced drug delivery systems like liposomes and nanoparticles. These systems are specifically designed to enhance the in vivo concentration and bioavailability of THC. Furthermore, the pursuit of THC-based Mannich base derivatives, which have demonstrated superior antioxidant and anti-cancer activities due to their structurally optimized nature, represents a promising trajectory for future investigation.

#### 2.8.2. Applications of Synthetic Biology

The synthesis of THC has been optimized through the auspices of synthetic biology techniques and microbial engineering, resulting in enhanced production efficiency and reduced costs. These advancements not only furnish a more robust supply of raw materials to support clinical applications but also facilitate the expedited development of THC and its derivatives, thereby broadening the scope of their potential therapeutic utility.

Utilizing Escherichia coli BL21 (DE3) as a host cell, researchers have successfully expressed NADPH-dependent curcumin convertase (CurA) to facilitate the efficient conversion of curcumin to THC via an in vitro cascade catalytic process. This approach leverages a method for the effective synthesis of THC through a cascade enzyme system based on the NADPH cofactor cycle. Furthermore, the metabolic pathway for the production of curcumin and THC was established, with optimizations in fermentation conditions and enhancements in NADPH supply. This has enabled the synthesis of curcumin and THC from ferulic acid, thereby expanding the capabilities of biotechnological production of these compounds.

In separate investigations, the utilization of a specialized microorganism, Rhodococcus sp., was instrumental in the biotransformation of curcumin into hexahydrocurcumin and octahydrocurcumin. This demonstrates that the biosynthesis of curcumin and its derivatives can be successfully accomplished through the auspices of microbial engineering, thereby showcasing the versatility and potential of biotechnological approaches in the production of bioactive compounds.

In summation, the trajectory of future THC research should be directed towards the identification of novel biological targets, the enhancement of clinical trial design, the innovation of new dosage forms, and the integration of synthetic biology. Through these avenues of inquiry, we can gain a deeper understanding of the pharmacological mechanisms of THC, enhance its clinical utility, and devise fresh strategies and techniques for the management of associated pathologies.

#### 2.8.3. Challenge

##### Low Bioavailability

Bioavailability is a critical pharmacokinetic parameter that refers to the rate and extent to which a drug reaches the systemic circulation from its site of administration, and it is a pivotal determinant of therapeutic efficacy. Research has indicated that THC, including its solid dispersion form, exhibits enhanced bioavailability in mice when compared to curcumin; however, there are still limitations that need to be addressed [99,100,101,102]. Curcumin and its derivatives are typically characterized by poor solubility and structural instability, which significantly contribute to their low bioavailability. To overcome these challenges and improve the bioavailability of THC, investigators have explored various strategies, including the development of alternative administration routes and the formulation of different dosage forms. Additionally, microbial transformation was employed to modify the structure of curcumin, thereby enhancing its bioavailability and optimizing its potential for therapeutic application [95,97,103].

##### Long-Term Toxicity

Despite the prowess of THC in displaying favorable pharmacological activity, investigations into its long-term toxicity remain relatively scarce. The assessment of a drug’s long-term safety profile necessitates prolonged clinical trials, which are frequently prohibitively expensive and time-intensive. Moreover, the latent toxicity of a pharmaceutical agent may be attributed to its metabolites, underscoring the critical importance of conducting comprehensive analyses of THC and its metabolic pathways to thoroughly evaluate its long-term safety [104].

##### Individual Difference

Individual differences encompass the variability in response to a drug among different individuals, which can be attributed to a multitude of factors including genetics, age, gender, and the status of liver and kidney function [105]. Such disparities in response to THC may significantly influence its therapeutic efficacy and safety, necessitating their consideration in clinical practice. For instance, variations in the absorption, distribution, metabolism, and excretion (ADME) of THC across individuals can result in divergent outcomes in terms of treatment effectiveness or the incidence of adverse reactions [106]. In synthesis, while THC has exhibited promising pharmacological activity, the existing body of research confronts challenges related to its limited bioavailability, the uncertainty of its long-term toxicity, and the pronounced individual differences in its response. Future investigations should focus on devising strategies to enhance THC’s bioavailability, conducting comprehensive long-term toxicity studies, and gaining a more profound understanding of its mechanisms of action across diverse populations to support its clinical application.

## 3. Conclusions

THC, as the primary metabolite of CUR, has garnered substantial attention for its notable pharmacological activities and therapeutic promise in recent years. Derived from the rhizome of turmeric, curcumin is a naturally occurring compound that has been traditionally employed to modulate a plethora of conditions, including disorders of glucose and lipid metabolism. However, the intrinsic low bioavailability of curcumin has posed a significant impediment to its clinical utility. Structural modification of curcumin into THC not only enhances its stability but also amplifies its biological potencies, such as antioxidant, anti-inflammatory, and anti-cancer properties [95,107]. Research findings indicate that THC surpasses curcumin in terms of antioxidant activity and gastrointestinal absorption, demonstrating a spectrum of biological activities that are superior to those of its parent compound [39]. Notably, THC was implicated in neuroprotection by mitigating the G1/S block in Alzheimer’s disease models and inhibiting the cell cycle of microglia. Additionally, THC was shown to augment the inhibitory effects of curcumin on breast cancer cells, highlighting its potential in the realm of anti-cancer therapeutics [59,78,108].

The exploration of THC extends beyond its direct pharmacological effects to encompass the design and synthesis of its derivatives. Through structural modifications, researchers have engineered a series of novel THC derivatives that exhibit enhanced anti-cancer and antioxidant activities compared to the parent molecule. These investigations chart new territories and methodologies for the continued development of THC and its derivatives [11,78,108].

In essence, THC, as a naturally occurring bioactive molecule, introduces novel horizons in the treatment of inflammation-related and neurodegenerative diseases. Its multifaceted mechanism of action, particularly its role in modulating pivotal inflammatory pathways and bolstering the intrinsic antioxidant defense system, has demonstrated efficacy in cancer prevention and neuroprotection. Future research endeavors and clinical trials should focus on refining the administration of THC to address its hydrophilicity challenges and limited bioavailability, potentially through the deployment of nanotechnology or carrier systems to ensure effective concentrations at the site of action. Concurrently, a thorough evaluation of the safety profile of THC for long-term use and its potential interactions with other pharmaceuticals is crucial for its transition from the laboratory setting to clinical application. With the resolution of these challenges, THC is poised to emerge as an innovative therapeutic agent, expanding the arsenal of options available for clinical interventions.

## Figures and Tables

**Figure 1 ijms-26-03561-f001:**
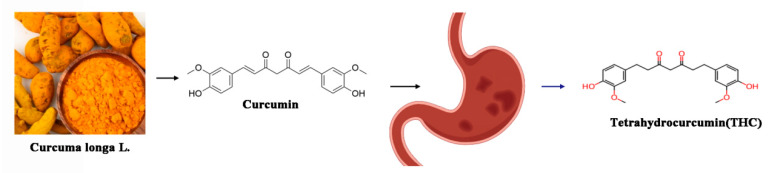
Source of tetrahydrocurcumin.

**Figure 2 ijms-26-03561-f002:**
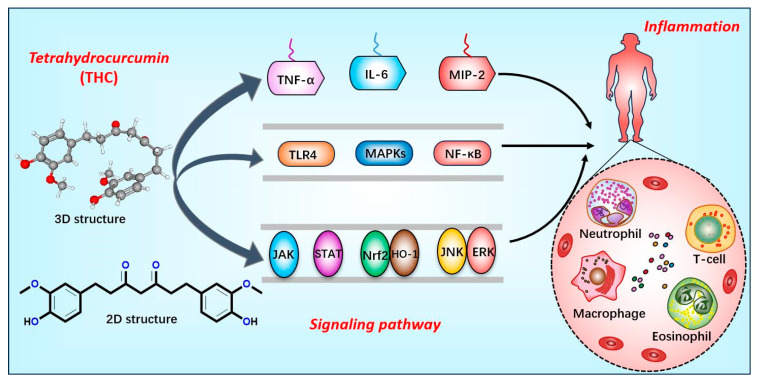
Relationship between tetrahydrocurcumin and inflammation. Tetrahydrocurcumin influences disease progression by regulating inflammatory cytokines (TNF-α, IL-6, MIP-2), key proteins (TLR4, MAPKs, and NF-κB), and signaling pathways (JAK/STAT, Nrf2/HO-1, and JNK/ERK).

**Figure 3 ijms-26-03561-f003:**
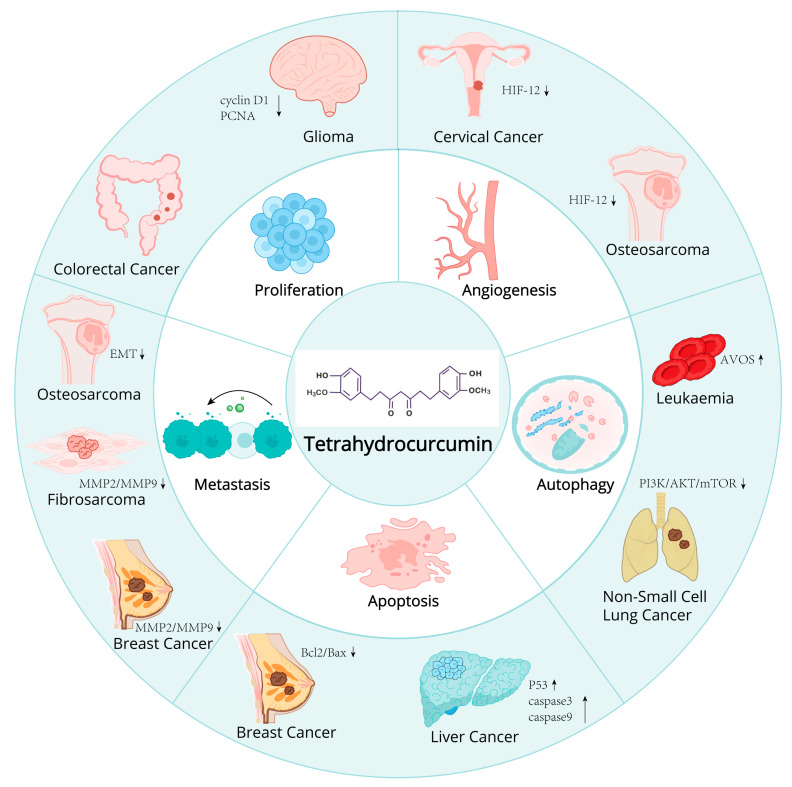
The antitumor mechanism of tetrahydrocurcumin and its role in various types of tumors.

**Figure 4 ijms-26-03561-f004:**
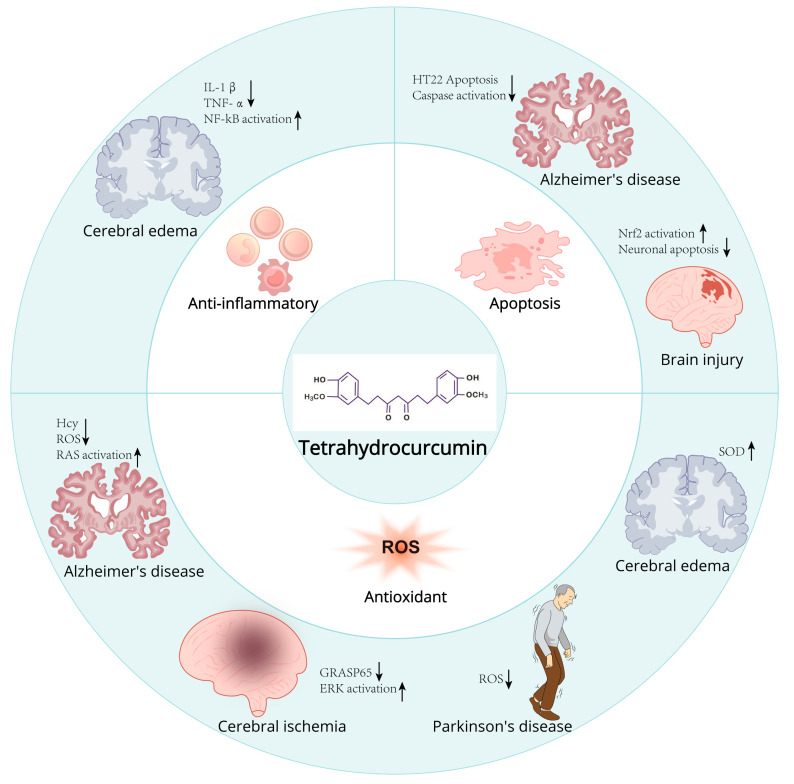
Mechanism and role of tetrahydrocurcumin in neurodegenerative diseases.

**Table 1 ijms-26-03561-t001:** Therapeutic impacts and molecular mechanisms of THC in cancer treatment.

Disease Name	Targets and Signaling Pathways	Effect	References
Cervical Cancer	Tumor angigenesis HIF-1-α	Suppressed tumor angiogenesis, volume, and growth rate	[63,64,65,66,67,68,69,70,71,72,73,74]
Breast Cancer	Bcl-2/Bax protein MMP-2 MMP-9 Apoptosis G2/M cell cycle arrest	Inhibited proliferation, facilitated apoptosis Diminished metastatic potential	[65,66,67]
Liver Cancer (HCC)	Caspase-3 Caspase-9 p53 gene Anti-angiogenic properties	Enhanced survival rate Suppressed proliferation Diminished ascites volume and abdominal girth	[63,68]
Acute Myeloid Leukemia	Apoptosis Autophagy AVOs	Elicited cell death in drug-resistant cells	[69,70]
Fibrosarcoma	MMP-2 MMP-9 Upa MT1-MMP TIMP-2	Reduced invasive and migratory capabilities Diminished cellular adhesion	[71]
Colon Cancer	BCL-2/BAX Caspase3 MMP-2 MMP-9 E-Cadherin N-Cadherin Vimentin	Inhibitted cell proliferation and transfer Promote apoptosis Inhibition of EMT transformation	[59,72]
Osteosarcoma	HIF-1α Akt/mTOR p38 MAPK signaling Autophagy MET process	Diminished proliferation, migration, and invasion Fostered MET Curbed angiogenesis	[73]
Non-Small Cell Lung Cancer (NSCLC)	Autophagy Beclin-1 LC3-II/LC3-I PI3K/Akt/mTOR signaling	Induced autophagy Suppressed growth and proliferation	[74]
Glioma	GSH levels Cyclin D1 PCNA expression G0/G1 cell cycle arrest	Augmented radiosensitivity Reduced tumor cell viability Increased apoptosis rate	[7,75]

**Table 2 ijms-26-03561-t002:** Therapeutic landscape of tetrahydrocurcumin in neurological conditions.

Disease Name	Targets and Signaling Pathways	Effect	References
Traumatic Brain Injury	Oxidative stress Mitochondrial dysfunction Nrf2 signaling pathway	Mitigates cerebral edema Reduces neuronal apoptosis	[80,81]
High-Altitude Cerebral Edema	IL-1β TNF-α SOD VEGF MMP-9 NF-κB	Reduces cerebral water content Diminishes IL-1β and TNF-α	[53,82]
Cerebral Ischemia	THcy levels Mitochondrial oxidative stress MMP-9 Tight junction proteins Autophagy markers ERK pathway	Enhances cerebral function Reduces infarct volume	[83,84,85,86]
Parkinson’s Disease	Dopaminergic neurons DA and DOPAC levels	Counteracts MPTP-induced depletion of DA and DOPAC	[87,88]
Alzheimer’s Disease	Aβ plaques Tau protein neurofibrillary tangles Ras signaling pathway Mitochondrial remodeling Oxidative stress Calcium ion influx Glutamate-induced injury	Mitigates Aβ-induced neurotoxicity Enhances learning/memory Enhances resilience to oxidative stress Reduces cell death	[72,89,90,91,92,93,94]

## Data Availability

There are no data associated with this study as it is a review article.

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
