# Peer review of "The Role of Tetrahydrocurcumin in Tumor and Neurodegenerative Diseases Through Anti-Inflammatory Effects"

_ijms, 2025, doi:10.3390/ijms26083561_

Round 1
Reviewer 1 Report
Comments and Suggestions for Authors
Dear Editor,
Anqi Zeng et al. present a systematic review on tetrahydrocurcumin (THC) and its potential therapeutic roles in anti-tumor effects and neurodegenerative diseases. The review is well structured and covers multiple perspectives
The following minor revisions should be addressed:
- NF-κB abbreviation appears on page 16, but it is not appropriately introduced. It should be defined upon its first mention in the text.
- Figure 4 has a similar illustration with a article that published in Sig Transduct Target Ther 6, 263 (2021). The figure should be redrawn to ensure originality. https://doi.org/10.1038/s41392-021-00658-5
- The synthetic scheme of THC by organic chemistry and its biosynthetic pathway from curcumin should be included in this review for a more comprehensive discussion.
Author Response
1.NF-κB abbreviation appears on page 16, but it is not appropriately introduced. It should be defined upon its first mention in the text.
Answer:Thank you for your question, we have described the full name of NF-KB under 2.5.3.2. I have marked in red in the article, thank you.
Figure 4 has a similar illustration with a article that published in Sig Transduct Target Ther 6, 263 (2021). The figure should be redrawn to ensure originality. https://doi.org/10.1038/s41392-021-00658-5
Answer:Thank you for your question.In combination with the editer and the core idea of this article, we have decided to delete this image. thank you.
The synthetic scheme of THC by organic chemistry and its biosynthetic pathway from curcumin should be included in this review for a more comprehensive discussion.
Answer:Thank you for your question.We will carry out more in-depth research on tetrahydrocurcumin.
Reviewer 2 Report
Comments and Suggestions for Authors
The manuscript is a narrative review on Tetrahydrocurcumin (THC), the main metabolite of curcumin, and its interest due to its anticancer and neuroprotective potential.
The manuscript is well written, makes reference to the mechanisms of action involved and the beneficial effects of THC observed in various studies, and I suggest it be accepted for publication after a minor review.
Suggestions:
- Please explain all abbreviations when they first appear in the text.
- In Figure 2 is MIF-2 written correctly?
- Briefly explain what each figure represents (the mechanisms or effects illustrated) and explain the abbreviations from figures in the figure's footer.
- In section 2.3.3 please complete by specifying the link between antioxidant effect and inflammation.
- Delete the sections 2.4.1. and 2.5.1. including figures 3 and 4, and 6 and 7, respectively (there are too many figures in the manuscript), which do not directly refer to THC and its mechanisms of action.
- In section 2.4.2.3, the following sentence is repeated : "Owing to its influence on lipid metabo-lism and modulation of food intake, THC may facilitate a reduction in visceral adiposity and body weight."
- In section 2.4.3.9. what kind of study is referred to? In vivo or in vitro? What are C6 cells? Please detail.
- Instead of: "2.5.2. Potential effects of THC on neuroprotection and repair" I suggest "2.5.2. Potential mechanisms of THC on neuroprotection and repair"
- La 2.5.2.1.: "Antioxygenation"??
- At 2.6.1.: Please rephrase more clearly what you mean by "a quintuple period"? Also: "a tiered dosing regimen of 20 mg/kg, 60 mg/kg, and 200 mg/kg" - are you referring to different groups? Please rephrase and explain clearly.
Author Response
- Please explain all abbreviations when they first appear in the text.
Answer:Thank you for your question.We have modified it and marked it in the article.
- In Figure 2 is MIF-2 written correctly?
Answer:Thank you for your question.Due to our mistake in drawing, MIP-2 was mistakenly written as MIIF-2, now it has been modified in the picture.Thank you.
- Briefly explain what each figure represents (the mechanisms or effects illustrated) and explain the abbreviations from figures in the figure's footer.
Answer:Thank you for your suggestion. We have modified and improved the article according to the requirements and won the red mark in the article. Thank you.
- In section 2.3.3 please complete by specifying the link between antioxidant effect and inflammation.
Answer:Thank you for your advice, your advice is very good and it really should be added a description of the association between inflammation and oxidation, which we have described in the paper and highlighted in red.
- Delete the sections 2.4.1. and 2.5.1. including figures 3 and 4, and 6 and 7, respectively (there are too many figures in the manuscript), which do not directly refer to THC and its mechanisms of action.
Answer:Thank you for your suggestion. These two parts are really not relevant to the main idea of this article. We have deleted them, including the relevant pictures. thank you
- In section 2.4.2.3, the following sentence is repeated : "Owing to its influence on lipid metabo-lism and modulation of food intake, THC may facilitate a reduction in visceral adiposity and body weight."
Answer:Thank you, we have removed the duplicate statements, thank you for your comments
- In section 2.4.3.9. what kind of study is referred to? In vivo or in vitro? What are C6 cells? Please detail.
Answer:Thank you for your question, but our description is not detailed enough, we have added the relevant description. thank you.
- Instead of: "2.5.2. Potential effects of THC on neuroprotection and repair" I suggest "2.5.2. Potential mechanisms of THC on neuroprotection and repair"
Answer:Thank you for your question.We have modified it according to your suggestion
- La 2.5.2.1.: "Antioxygenation"??
Answer:Thanks for your suggestion, we have modified it to Antioxidation. thank you.
- At 2.6.1.: Please rephrase more clearly what you mean by "a quintuple period"? Also: "a tiered dosing regimen of 20 mg/kg, 60 mg/kg, and 200 mg/kg" - are you referring to different groups? Please rephrase and explain clearly.
Answer:Thanks for your suggestion, We did not make it clear in the paper. The specific administration method is once a day within 30 days, which has been corrected and marked in red in the paper, thank you.